# Testing for Thrombophilia in Young Cryptogenic Stroke Patients: Does the Presence of Patent Foramen Ovale Make a Difference?

**DOI:** 10.3390/medicina58081056

**Published:** 2022-08-05

**Authors:** Mantas Jokubaitis, Rūta Mineikytė, Lina Kryžauskaitė, Lina Gumbienė, Lina Kaplerienė, Saulius Andruškevičius, Kristina Ryliškienė

**Affiliations:** 1Centre of Neurology, Institute of Clinical Medicine, Faculty of Medicine, Vilnius University, LT-08661 Vilnius, Lithuania; 2Hematology, Oncology and Transfusion Medicine Center, Vilnius University Hospital Santaros Klinikos, LT-08661 Vilnius, Lithuania; 3Clinic of Cardiovascular Diseases, Institute of Clinical Medicine, Faculty of Medicine, Vilnius University, LT-08661 Vilnius, Lithuania; 4Center of Neurology, Center of Anesthesiology, Intensive Care and Pain Management, Vilnius University Hospital Santaros Klinikos, LT-08661 Vilnius, Lithuania

**Keywords:** stroke, patent foramen ovale, thrombophilia, antiphospholipid syndrome, leiden mutation, prothrombin mutation, hyperhomocysteinemia, family history

## Abstract

*Background and Objectives*: The diagnostic value of thrombophilia remains unknown in young patients with patent foramen ovale (PFO) and stroke. In this study we hypothesized that inherited thrombophilias that lead to venous thrombosis are more prevalent in patients with PFO. *Materials and Methods*: The study included patients of the tertiary center Vilnius University Hospital Santaros Klinikos who had a cryptogenic ischemic stroke between the ages of 18 and 50 between the years 2008 and 2021. Transient ischemic attacks were excluded. Contrast-enhanced transcranial Doppler ultrasound and extensive laboratory testing were performed. *Results*: The study included 161 cryptogenic stroke patients (mean age 39.2 ± 7.6 years; 54% female), and a right-to-left shunt was found in 112 (69.6%). The mean time between stroke and thrombophilia testing was 210 days (median 98 days). In total, 61 (39.8%) patients were diagnosed with thrombophilia. The most common finding was hyperhomocysteinemia (26.7%), 14.3% of which were genetically confirmed. Two patients (1.2%) were diagnosed with factor V Leiden mutation, three patients (1.9%) with prothrombin G20210A mutation, one patient (0.6%) had a protein C mutation and one patient (0.6%) had a protein S mutation. No antithrombin mutations were diagnosed in our study population. A total of 45.5% of patients with inherited thrombophilia had a right-to-left shunt, while 54.5% did not, *p =* 0.092. Personal thrombosis anamnesis was positive significantly more often in patients with antiphospholipid syndrome. *Conclusions*: The hypothesis of the study was rejected since inherited venous thrombophilia was not significantly more common in patients with PFO. Due to the rarity of thrombophilias in general, more research with a larger sample size is required to further verify our findings.

## 1. Introduction

Despite rigorous diagnostic workup, about one-third of all acute ischemic strokes are identified as cryptogenic [1]. Numerous studies have confirmed an association between patent foramen ovale (PFO) and cryptogenic stroke, particularly in younger individuals without traditional cardiovascular risk factors such as hypertension, dyslipidemia, and diabetes mellitus [2]. However, due to the high prevalence of PFO in healthy subjects [3], it is necessary to explore whether the PFO is not an incidental finding and whether there is another mechanism of stroke such as extracranial or intracranial arterial pathology, paroxysmal atrial fibrillation (AF), thrombophilia or other causes [4,5,6].

At first sight, testing for inherited or acquired thrombophilias may shed light on the pathophysiology of a thrombotic event and have an impact on the strategy for secondary stroke prevention in patients with PFO. However, thrombophilia testing in ischemic stroke is expensive and not routinely recommended even in the young [7,8]. The current absence of robust evidence indicating a significant association between PFO, thrombophilia, and ischemic stroke as identified by thrombophilia testing calls the rationale for such evaluation into question.

Thrombophilia prevalence varies: in Whites, factor V Leiden mutation accounts for 5% and prothrombin mutation accounts for 3%; lupus anticoagulants are found in up to 5% of individuals, whereas decreased protein C, protein S, and antithrombin activity occurs in less than 0.5% of cases [9]. The risk of initial venous or arterial thromboembolism depends on the type of thrombophilia–heterozygous factor V Leiden, prothrombin mutations increase the risk of venous thromboembolism up to four-fold (the risk being higher in a homozygous or compound state), antithrombin deficiency up to 16-fold, protein C and S deficiency up to 8-fold and 7-fold respectively, whereas the presence of antiphospholipid antibodies are associated with both arterial (primarily) and venous thrombosis risk increase [10]. In addition, the role of hyperhomocysteinemia and the increased concentration of lipoprotein a is less clear. Studies have demonstrated that hyperhomocysteinemia at levels above 22 µmol/L can raise the risk of venous thromboembolism four-fold [11], whereas for lipoprotein a level higher than 63 nmol/L may result in up to a two-fold risk increase of venous thromboembolism in adults [12].

It must be stressed that the timing of thrombophilia testing is critical. Due to various underlying medical conditions (such as acute thrombosis, pregnancy, inflammation, etc.) thrombophilia testing may yield falsely low levels of protein C, protein S and antithrombin as a consequence of the consumption of these factors. In addition, the use of anticoagulants, which are warranted in the case of acute venous thromboembolism, can result in false positive results, particularly in terms of the antiphospholipid antibodies test [13]. Therefore, testing for the aforementioned thrombophilias within three months of a thrombotic event may result in higher medicals cost due to the need for test repetition. Furthermore, false positive results can lead to the incorrect diagnosis of a factor deficiency, whereas normal results can provide false reassurance.

Thrombophilias are usually classified based on etiology (acquired/inherited) and the risk of venous or arterial thrombosis. This is a clinically valuable classification since the extent of thrombophilia testing can be determined based on the results of contrast-enhanced transcranial Doppler ultrasound (c-TCD) and/or contrast-enhanced transesophageal echocardiography, i.e., in case of cryptogenic stroke, if PFO is not found, only the tests for arterial thrombosis should be ordered, whereas if the presence of PFO is confirmed, both arterial and venous thrombosis tests should be performed. However, such categorization could be misleading since there is evidence that hereditary thrombophilias have a role in arterial thrombosis [14].

In this study, we hypothesized that inherited thrombophilias that lead to venous thrombosis is more prevalent in patients with PFO.

## 2. Materials and Methods

Patients with cryptogenic ischemic stroke occurring between 18–50 years of age during 1 January 2008 to 4 March 2021 at the Vilnius University Hospital Santaros Klinikos were considered for inclusion of this retrospective study. The cryptogenic origin was defined using Embolic Strokes of Undetermined Source (ESUS) [1] criteria i.e., non-lacunar stroke, without proximal arterial stenosis or cardioembolic sources (computed tomography (CT) and/or magnetic resonance tomography (MRI) was performed to visualize the stroke, CT/MRI angiography, extracranial artery ultrasound and/or digital subtraction angiography was done to exclude arterial stenosis, transthoracic and/or transesophageal ultrasound, electrocardiography and 24-h Holter monitoring was performed to exclude possible cardioembolic sources). Transient ischemic attacks were not included in this study. The cryptogenic stroke patients were then examined by the same physician performing c-TCD following the Venice consensus [15] i.e., the contrast agent was prepared using 8 mL isotonic saline solution, 1 mL air, and 1 mL patient’s blood mixture and injected as a bolus. During c-TCD monitoring, the shunt was evaluated at rest and after the Valsalva maneuver. The patients were tested for antiphospholipid antibodies (anti-β_2_ glycoprotein-1 IgGAM, if positive, additionally tested for IgA, IgM, IgG; anti-cardiolipin IgGAM, if positive, additionally tested for IgA, IgM, IgG; lupus anticoagulant–dilute Russel Viper Venom Time (dRVVT) screening test and ratio, dRVVT confirmation test and ratio, normalization ratio; if any of the antiphospholipid antibodies were positive, tests were repeated after 12 weeks and only then classified as a true positive), homocysteine (normal range: female 4.44–13.56 μmol/L; male 5.46–16.20 μmol/L), lipoprotein a (normal range: <75 nmol/L), protein C (normal range: 54–166%), protein S (normal range: female 50–134%; male 74–120%), factor VIII (normal range: 52–290%), antithrombin (normal range: 66–124%) levels, factor V Leiden and prothrombin mutations. The aforementioned tests were carried out (or non-genetic tests were re-done in case of testing at <3 months) at least three months after the ischemic event. A group of venous thromboembolism patients was identified as having at least one of the following mutations: the increased activity of factor VIII, prothrombin mutation, Leiden V mutation, protein C or protein S deficit. In addition, all patients were tested for Fabry disease, and the majority of patients were tested for syphilis, human immunodeficiency virus. Finally, lumbar puncture and/or urine drug tests were performed depending on the clinical circumstances. All of the patients were asked about their personal or family history of thrombotic events. Females were additionally asked about their history of abnormal pregnancies.

A statistical analysis was completed using SPSS (Statistical Package for the Social Sciences) version 26.0. All included patients were divided into two groups: with or without detected shunt during the c-TCD testing. A chi-squared test was used to compare categorical variables. For small-sized samples, Fisher’s exact test was used. The results were considered as statistically significant when the *p* value was less than 0.05. Qualitative data were provided in numbers and percentages.

Ethics approval number 2019/2-1099-579 was issued by the Vilnius Regional Biomedical Research Ethics Committee on 26 February 2019 and was extended on 23 March 2021.

## 3. Results

The study included a total of 161 patients that met ESUS criteria for cryptogenic stroke (mean age 39.2 ± 7.6 years). Fifty-four percent of patients were female (mean age 39.8 ± 6.8 years) and 46% were male (mean age 38.6 ± 8.5 years). There was no significant age difference between males and females. The mean time between stroke and thrombophilia testing was 210 days (median 98 days). In total, 64 (39.8%) patients were diagnosed with thrombophilia. The thrombophilia tests performed are shown in Table 1. In 84 (52.2%) patients, all of the aforementioned tests (excluding Lipoprotein a) were performed. Hyperhomocysteinemia (17.64 ± 9.06 μmol, min 12.10 μmol/L, max 64.00 μmol/L) was the most common finding. Hyperhomocysteinemia ranges were set as follows: low 12–30 μmol/L–40 (93.0%), moderate 30–100 μmol/L–3 (7.0%), severe > 100 μmol/L–0 cases. A family history of thrombotic events in first degree relatives was positive in 25 patients, whereas a previous thrombotic event had occurred in 31 patients. Personal thrombosis anamnesis was positive significantly more often in patients with antiphospholipid syndrome (*p =* 0.013). Additional data regarding the personal and family history of thrombosis are shown in Table 2.

Almost seventy percent of study patients had a right-to-left shunt, with 48.4% (of the total) having a shunt detected at rest. Any shunt (at rest or during the Valsalva maneuver) was observed in 45 (70.3%, *p =* 0.867) of the patients with thrombophilia, whereas a shunt at rest was found in 29 (45.3%, *p =* 0.579) thrombophilia patients. Correlations between the detected shunt during the c-TCD testing and inherited venous thrombophilias are shown in Table 3.

## 4. Discussion

The hypothesis that inherited thrombophilias, which cause venous thrombosis, are more common in PFO patients, was rejected. Nevertheless, due to the rarity of thrombophilia in general, more research with a larger sample size is required to further verify our findings.

Hyperhomocysteinemia was the most common finding in our study. It is known that elevated levels of homocysteine are associated with the increased risk of stroke via multiple mechanisms such as increased reactive oxygen species, endothelial injury and the promotion of inflammation [16]. However, it is not yet clear what level of hyperhomocysteinemia is clinically significant. In addition, testing for homocysteine levels in blood remains controversial because no homocysteine-lowering therapy has been shown to reduce the risk of thrombotic events [8,17]. Consequently, testing for hyperhomocysteinemia appears to be futile in the absence of effective therapy, demanding more research in the pursuit of stroke risk reduction.

Testing for acquired antiphospholipid syndrome may be considered in the presence of a history of prior venous thromboembolism, second trimester abortion, or rheumatologic disorder, as recommended by American Stroke Association (ASA) guidelines [8]. However, we suggest that testing for thrombophilia that could be potentially treated with anticoagulants may benefit young cryptogenic stroke patients. Our finding of antiphospholipid syndrome prevalence in patients with a prior personal thrombotic event could be one of the arguments for this suggestion. Therefore, testing for antiphospholipid antibodies is reasonable because the findings may necessitate lifelong anticoagulant therapy and may defer PFO closure [6].

Testing for inherited thrombophilia in patients with cryptogenic stroke is controversial because the results do not affect the standard antiplatelet approach. Antiplatelet therapy is initiated in situations with verified prothrombin and factor V Leiden mutations, as well as reduced protein C, S, antithrombin levels, and increased FVIII activity, further raising the question of the futility of such testing. Moreover, ASA 2021 guidelines recommend testing of only carefully selected patients, but the selection criteria are poorly defined, e.g., it is unclear whether testing should be based on the presence of pregnancy-related disorders, a personal history of thrombosis, or, in the case of suspected inherited thrombophilia, a positive family history [8]. Additionally, in spite of the availability of advanced imaging and testing, some authors have acknowledged the issue of unsatisfactorily high rates of cryptogenic stroke [18]. As a result, the need for a stricter ESUS definition has emerged. The mnemonic AHEAD has been proposed to represent a wide range of testing modalities prior to ESUS diagnosis, with the letter D standing for differential diagnosis, which includes, among other tests, laboratory assessment for inherited thrombophilias. According to the authors, a stricter ESUS definition and extended testing may aid in the discovery of stroke etiology and, as a result, maximize risk reduction of stroke recurrence with appropriate treatment [18].

Our findings call into question the utility of categorizing patients with thrombophilia based on their risk of venous and arterial thrombosis in the setting of PFO. The frequency of thrombophilia did not differ significantly between the groups with and without PFO, e.g., there were patients with cryptogenic stroke, no PFO, but a positive diagnosis of venous thrombophilia. Therefore, this study puts into consideration the role of commonly recognized “venous thrombophilia” in arterial thrombosis. We emphasize that in young stroke patients without traditional cardiovascular risk factors, with or without PFO, and even without family history, that it is reasonable to test for inherited thrombophilias. Although a previous meta-analysis has found a relationship between hereditary thrombophilia and an elevated risk of arterial ischemic stroke [14], this link has to be verified in patients specifically evaluated for PFO in adequately powered multicenter studies.

The strengths of our study include the extensive laboratory testing of study participants and subgroup analysis based on foramen ovale status. Nevertheless, there are few limitations associated with our study. Firstly, only half of the patients had been tested for all thrombophilia. This is due to our center’s financial and technical circumstances changing over time, as well as the identification of new thrombophilia during the study period. Therefore, only “arterial thrombophilias” were tested in the early years of the study, with later additions of inherited thrombophilia markers and, most recently, Lp (a) and FVIII tests. A relatively low sample size prevented us from sophisticated statistical analysis. Moreover, because not all patients had transesophageal echocardiography done, anatomical aspects of right-to-left shunt remained unstudied.

## 5. Conclusions

Thrombophilia prevalence was low even among strictly selected cryptogenic stroke patients. The most frequent finding was hyperhomocysteinemia. Although rare, inherited venous thrombophilia was found to be equally frequent in individuals with and without PFO. Inherited thrombophilia, which is generally perceived to be venous, might also influence the development of arterial thrombosis. As a result, even in the absence of PFO, the testing of inherited thrombophilia should be considered in young ischemic cryptogenic stroke patients.

## Figures and Tables

**Table 1 medicina-58-01056-t001:** Number of laboratory tests performed and thrombophilia diagnosis.

Laboratory Test	Number of Patients Tested, *n* (%)	Thrombophilia Diagnosis, *n*
Antiphospholipid antibodies	154 (95.7%)	6 **
Homocysteine	139 (86.3%)	43 ***
Factor V Leiden mutation	131 (81.4%)	2 ****
Prothrombin mutation *	131 (81.4%)	3 ****
Factor VIII activity	122 (75.8%)	4
Antithrombin	121 (75.2%)	-
Protein C	116 (72.1%)	1
Protein S	113 (70.2%)	1
Lipoprotein a	49 (30.4%)	4

* G20210A mutation; ** Antiphospholipid syndrome; *** 22 cases of elevated homocysteine level, 21 cases confirmed by genetic testing (11 homozygous, 10 heterozygous); **** Heterozygous mutation.

**Table 2 medicina-58-01056-t002:** Association of personal and family history and thrombophilia diagnosis.

	APS (+), *n* = 6	APS (-)	*p*	PTM or FVL (+), *n* = 5	PTM or FVL (-)	*p*	HHcy (+), *n* = 43	HHcy (-)	*p*	MTHFR (+), *n* = 21	MTHFR (-)	*p*
Positive personal history of thrombosis	4 (66.7%)	33 (21.3%)	**0.025**	1 (20%)	30 (19.2%)	1	6 (14.0%)	31 (26.3%)	0.100	3 (14.3%)	34 (24.3%)	0.411
Positive family history of thrombosis	2 (33.3%)	23 (14.7%)	0.234	2 (40%)	23 (14.7%)	0.172	6 (14.0%)	19 (16.1%)	0.739	3 (14.3%)	22 (15.7%)	1
Positive personal or family history of thrombosis	4 (66.7%)	51 (32.9%)	0.182	3 (60%)	52 (33.3%)	0.339	12 (27.9%)	43 (36.4%)	0.352	6 (28.6%)	49 (35.0%)	0.630

APS—antiphospholipid syndrome; PTM—prothrombin G20210A mutation; FVL—factor V Leiden mutation; HHcy—hyperhomocysteinemia; MTHFR—methylenetetrahydrofolate reductase (homozygous and heterozygous mutations).

**Table 3 medicina-58-01056-t003:** Relation of contrast-enhanced transcranial Doppler ultrasound results with inherited venous thrombophilia.

Thrombophilia	Shunt (-)	Shunt (+)	*p*	Shunt at Rest (-)	Shunt at Rest (+)	*p*
Increased activity of FVIII	2	2	0.281	2	2	1
Prothrombin mutation	2	1	0.165	3	0	0.101
Leiden V mutation	2	0	0.066	2	0	0.218
Protein C deficit	0	1	1	1	0	0.400
Protein S deficit	0	1	1	0	1	1
Total	6	5	0.092	8	3	0.140

## Data Availability

The anonymized data presented in this study are available upon reasonable request from the corresponding author.

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
