# Peer review of "Testing for Thrombophilia in Young Cryptogenic Stroke Patients: Does the Presence of Patent Foramen Ovale Make a Difference?"

_medicina, 2022, doi:10.3390/medicina58081056_

Round 1

Reviewer 1 Report

The manuscript is written in good manner. The text is well structured,  all parts of text are clear.  The methods are described clearly and used methods are appropriate to study. Conclusions are relevant to results of study. 

As authors note the one of the main limitation is small study population and retrospective nature of it.  On the other hand, thrombophylia is rare condition in clinical practice,  so to reach the required number of patient in one center could take long time.

Some minor comments are presented below:

Line 73: please use CI unit (change mg/l to mmol/l)

The patient with antiphospholipide antibodies were not included in correlation analysis (Table 3). Could You please the reasons of exclusion of these patients or add them to analysis? 

Reviewer 2 Report

Interesting topic, of value to stroke doctors and patients.  A few items to address: 

1. most of us do not check homocysteine because, as you mention, "no homocysteine-lowering therapy has been shown to reduce the risk of thrombotic events."  Therefore, please explain why this was included. Also, if you were to exclude it, would the results be the same?  Would inherited venous thrombophilia still be equally frequent in individuals with and without PFO?

2. A limitation in this study is that not 100% of subjects were tested for the same inherited thrombophilias, and some as few as 30-70%.  This may not capture all the positive patients (excluded inadvertently).  You mention that limitation in the discussion, but would like more detail.  Why only half of the patients were tested?

3.  Another limitation in this study is that you are missing a few other inherited thrombophilias.  Currently, at our hospital, and at colleagues hospitals, we test for about 11-13 thrombophilias (excluding homocysteine).  Would mention that in discussion and why other tests were not performed.  Perhaps they do not have as much value because they are even more rare and more expensive to order?

4. In discussion, it is awkward to ask a question, so would rephrase this sentence: "e.g., should testing be based on the presence of pregnancy-related disorders, personal history of thrombosis or, in the case of suspected inherited thrombophilia, should positive family history be a determinant for additional evaluation?"

5. Because it is reasonable to test for thromophilias in young patients even without family history, this phrase is incomplete: "We emphasize that in patients without a family history of thrombotic events, routine thrombophilia testing should be avoided since conventional stroke risk factors are far more frequent than inherited thrombophilia."   Please rephrase.

6. The idea of "lifelong anticoagulant therapy and defer from PFO closure" is a bit overstated.  One could imagine a situation where PFO would be closed in recurrent events on AC, especially if PFO caused cardiac symptoms or was very large.  Therefore, would add "may defer PFO closure" rather than "defer from" to clarify.   

7. It is great that you bring up this relevant topic of how do consider a more thorough review/workup before considering the diagnosis of ESUS on these patients.  You may want to include this paper in the discussion that describes a mnemonic called AHEAD in which the D in AHEAD refers to Differential diagnosis that includes laboratory testing for inherited thrombophilias -- many ESUS trials did not include it, but I agree with you that it is needed.  Here is the reference: "Forge AHEAD with stricter criteria in future trials of embolic stroke of undetermined source." Neural Regen Res. 2022 May;17(5):1009-1010. doi: 10.4103/1673-5374.324838. PMID: 34558523; PMCID: PMC8552872.

8.  The optimal terminology for hypothesis testing is accepted or rejected, so perhaps you may want to change "disproved" to "rejected."

9. in the introduction, would change "accidental" to "incidental"

10. Can you explain in a bit more detail this "false reassurance" phenomenon at 3 months please?  Why would use of AC causing false positive results of APLA (reference 13) be reassuring 3 months later when retested?

11.  Because PFO testing modality is variable, this sentence is incomplete: "in case of cryptogenic stroke if PFO is not found, only the tests considering arterial thrombosis should be ordered, whereas if PFO presence is confirmed, both arterial and venous thrombosis tests should be performed."  As you know, TTE alone misses some PFOs, so we would typically also use TEE with bubble study confirming no PFO before not ordering thrombophilias.  Perhaps clarify this point.

Thank you and great work!
